# The Dual Role of Chemerin in Lung Diseases

**DOI:** 10.3390/cells13020171

**Published:** 2024-01-16

**Authors:** Philomène Lavis, Benjamin Bondue, Alessandra Kupper Cardozo

**Affiliations:** 1Department of Pathology, Brussels University Hospital, Université Libre de Bruxelles, 1070 Brussels, Belgium; philomene.lavis@ulb.be; 2Institut de Recherche Interdisciplinaire en Biologie Humaine et Moléculaire (I.R.I.B.H.M.), Université Libre de Bruxelles, 1070 Brussels, Belgium; benjamin.bondue@ulb.be; 3Department of Pneumology, Hôpital Universitaire de Bruxelles, Université Libre de Bruxelles, 1070 Brussels, Belgium; 4Inflammation and Cell Death Signalling Group, Signal Transduction and Metabolism Laboratory, Université Libre de Bruxelles, 1070 Brussels, Belgium

**Keywords:** chemerin, RARRES2, CMKLR1, lung, inflammation

## Abstract

Chemerin is an atypical chemokine first described as a chemoattractant agent for monocytes, natural killer cells, plasmacytoid and myeloid dendritic cells, through interaction with its main receptor, the G protein-coupled receptor chemokine-like receptor 1 (CMKLR1). Chemerin has been studied in various lung disease models, showing both pro- and anti-inflammatory properties. Given the incidence and burden of inflammatory lung diseases from diverse origins (infectious, autoimmune, age-related, etc.), chemerin has emerged as an interesting therapeutical target due to its immunomodulatory role. However, as highlighted by this review, further research efforts to elucidate the mechanisms governing chemerin’s dual pro- and anti-inflammatory characteristics are urgently needed. Moreover, although a growing body of evidence suggests chemerin as a potential biomarker for the diagnosis and/or prognosis of inflammatory lung diseases, this review underscores the necessity for standardizing both sampling types and measurement techniques before drawing definitive conclusions.

## 1. Introduction

Both chronic and acute lung inflammation are a major public health burden [1]. Etiologies are various, and adequate therapies are needed to restore homeostasis. The lungs are in direct contact with external components, indicating that a quick and appropriate inflammatory response needs to be initiated to maintain the integrity of the system [2]. This is made possible thanks to the coordinated action of the immune system, an integrated network of organs, cells and proteic mediators (cytokines and chemokines). However, a disproportionate acute immune response can cause as much tissue damage as the initial trigger. This was recently observed during the COVID-19 pandemic, where the development of acute respiratory distress syndrome (ARDS) was not linked to the viral load but to a deregulated inflammatory response [3]. The resolution of inflammation is also as important as an appropriate initial immune response. Indeed, the persistence of immune cells and pro-inflammatory cytokines can lead to tissue damage, fibrosis and the alteration of organ function [4].

Chemerin is an atypical chemokine first identified about 20 years ago and described as a potent chemoattractant agent for some immune cells [5,6]. However, in murine models of pulmonary diseases, chemerin displayed pro- and anti-inflammatory properties, suggesting a fine and complex immunodulatory role for this protein (Figure 1). Accordingly, this review aims to describe and discuss data on the diverse roles of chemerin in lung inflammation.

## 2. Chemerin and Its Receptors

Preprochemerin is a small protein of 163 amino acids encoded by the gene retinoic acid receptor responder 2 (*RARRES2*), and it is secreted by a wide range of cells, mainly adipocytes, hepatocytes and fibroblasts [5,7,8,9,10]. The cleavage of the 20-amino acid signal peptide from preprochemerin leads to prochemerin, the most abundant isoform of the protein found in the basal state. This preprotein binds with low affinity to its receptors and does not lead to cell activation [6,11]. A cleavage of six to seven amino acids from its C-terminal extremity is required to obtain the active form, namely chemerin [12]. The most active forms of chemerin (Chemerin 21-157 and chemerin 21-156) are generated through the proteolytic action of two neutrophil serine proteases, neutrophil elastase and cathepsin G, respectively [13]. Chemerin 21-157 can also result from a cleavage by cathepsins L and K and carboxypeptidases N and B [14,15]. However, some proteases can also generate inactive forms of chemerin. Mast cell chymase is responsible for the removal of two to three amino acids from bioactive chemerin forms, and neutrophil proteinase 3 generates chemerin 21-155 [16]. Chemerin 21-158 is obtained through the proteolytic action of plasmin and tryptase [17]. The various isoforms of chemerin are summarized in Figure 2.

Chemerin was first described as the natural ligand of the G protein-coupled receptor chemokine-like receptor 1 (CMKLR1), also named ChemR23 [6]. The binding of chemerin to CMKLR1 leads to the activation of the ERK1/2 and PI3K/Akt pathways, as well as the activation of phospholipase C and calcium release [6,18]. Beta-arrestins are also recruited, leading to the internalization of CMKLR1 [19]. The chemerin/CMKLR1 axis was first described as a major chemoattractant system for monocytes, immature plasmacytoid and myeloid dendritic cells (DC) and natural killer (NK) cells [6,14,20,21,22,23]. CMKLR1 is, however, not expressed by neutrophils and lymphocytes, and therefore the system has no chemoattractant effect on these cells [24]. In addition to immune cells, CMKLR1 is expressed by fibroblasts, adipocytes, endothelial cells and pericytes, and the chemerin/CMKLR1 system has been shown to modulate angiogenesis and adipogenesis [8,10,25]. The second receptor described for chemerin is G protein-coupled receptor 1 (GPR1). In contrast to CMKLR1, GPR1 is only expressed by non-immune cells [26]. The affinity of chemerin for GPR1 is higher than for CMKLR1, but it only leads to a weak signaling, even if beta-arrestins are recruited and allow an internalization of the receptor [19]. The last chemerin receptor described is chemokine CC motif receptor-like 2 (CCRL2). Chemerin has the least affinity for this receptor, and its binding does not lead to cell activation or internalization [19]. It has then been proposed that CCRL2 is a decoy receptor, concentrating chemerin in sites of inflammation to facilitate its binding to CMKLR1 [27,28].

## 3. Chemerin and Lung Inflammation

### 3.1. Chemerin and Acute Respiratory Distress Syndrome

Acute respiratory distress syndrome (ARDS) is an acute inflammation of the lung linked to a sudden increase in alveolocapillary permeability, leading to decreased lung compliance, respiratory dysfunction and hypoxemia [29]. The diagnosis of ARDS is based on the Berlin criteria: 1. Development of an acute hypoxemia within 1 week of a known injury or new onset or worsening of respiratory symptoms. 2. Bilateral lung opacities on chest imaging. 3. Absence of a cardiac cause [30]. The severity of arterial hypoxemia determines the severity of ARDS in mild (PaO_2_/FIO_2_ ≤ 300 mm Hg), moderate (PaO_2_/FIO_2_ ≤ 200 mm Hg) and severe (PaO_2_/FIO_2_ ≤ 100 mm Hg). ARDS is responsible for 10% of intensive care unit admissions, and mortality approaches 50% in severe cases [31]. Etiologies are multiple, including either direct damage to the lung, mostly by viral or bacterial pneumonia, or an indirect lesion, notably secondary to sepsis or a severe trauma.

The most commonly used mouse model of ARDS is the acute lung injury (ALI) model that consists in the administration of bacterial lipopolysaccharide (LPS) [32]. This leads to a massive infiltration of neutrophils which peaks after 24 h, a development of edema and an intra-alveolar hemorrhage [33,34]. This model is considered as a model of mild ARDS [30]. Our group has previously shown in the direct ALI model induced by LPS that the addition of recombinant chemerin to LPS led to a decreased immune response characterized by a lower recruitment of neutrophils in the lungs and bronchoalveolar lavage fluid (BALF) and resulting in fewer histological lesions [35]. Pro-inflammatory cytokines were also decreased in the BALF of mice receiving recombinant chemerin and LPS. As expected, CMKLR1 knock-out (CMKLR1^KO^) mice did not respond to recombinant chemerin and presented higher levels of neutrophils in the lungs and BALF compared to WT mice after LPS challenge [35]. Please see Table 1; this table summarizes the main studies discussed in this review.

A recent study by Mannes et al. used an agonist of CMKLR1 coupled with ^64^Cu as a radiotracer to visualize the recruitment of CMKLR1-positive cells with positron emission tomography (PET). They observed a higher and significant uptake of the radiotracer in LPS-treated mice compared to control mice at days 1, 2, 4 and after LPS intratracheal instillation, with a maximal uptake at day 2. This increase was mostly linked to an uptake by monocytes-derived macrophages and to a lesser extent of interstitial macrophages and monocytes [36]. These results are compatible with the previously described chemoattractant role of chemerin in the LPS model of direct ALI.

In contrast to the results described above, Provoost et al. showed a pro-inflammatory role of the chemerin/CMKLR1 system in another model of acute lung inflammation induced by exposing the mice to diesel exhaust particles (DEP). They first observed that chemerin concentration in BALF was increased in WT mice exposed to DEP compared to control mice. This rise was associated with a decrease in the expression of chemerin in alveolar epithelial cells, suggesting a release of chemerin by these cells in the case of DEP exposure. CMKLR1^KO^ mice exposed to DEP presented a lower recruitment of monocytes and dendritic cells, as well as reduced levels of pro-inflammatory cytokines [37].

Another model of direct ALI is with exposure to ozone (O_3_) [38]. Razvi et al. exposed wild-type mice to O_3_ for 3 h and collected BALF 24 h later. Higher chemerin concentrations were observed in O_3_-exposed mice compared to control mice [39]. This model was then applied to CCRL2^KO^ mice and, although they presented the same degree of severity as WT mice, they had significantly higher chemerin concentrations in their BALF, confirming the role of CCRL2 as a decoy receptor [40]. These results indicate that CCRL2 has no evident role in the effects of chemerin during ARDS. Moreover, the fact that increased chemerin levels in the CCRL2^KO^ mice did not change the responses to O_3_ exposure may indicate that the maximum dose effect of chemerin was already reached in the WT mice.

Zou et al. used a model of indirect ALI in rats induced by limb ischemia/reperfusion. They observed an increase in the levels of chemerin in the lungs of rats as compared to controls. Rats treated with hydrogen-saturated physiological saline solution, a fluid showing protective lung effects in the case of ischemia/reperfusion, presented a decrease in chemerin lung levels that was strongly correlated with a decreased lung injury score as evaluated histologically [41].

All these studies highlight that the chemerin/CMKLR1 system is associated with acute inflammation, with higher chemerin levels measured in rodents with ALI [37,39,41] and an infiltration of CMKLR1-positive cells in the latter phases of the LPS model [36]. However, models of direct ALI induced by either LPS or DEP showed contradictory results, with an anti-inflammatory role for the first model [35] and a pro-inflammatory one for the second [37]. Even if LPS and DEP both activate the innate immune system through interaction with toll-like receptor 4, the intensity of the immune response is different. Indeed, the total number of neutrophils in the BALF is more than twice as high in mice receiving LPS as mice exposed to DEP [35,37,42,43]. Intratracheal instillation of chemerin reduced the inflammation induced by LPS instillation, possibly favoring the recruitment of a cell population with anti-inflammatory properties [35].

### 3.2. Chemerin, Lung Infection and Sepsis

As the chemerin/CMKLR1 system is involved in the physiopathology of lung acute inflammation, it makes sense that it could also be involved in lung infection and sepsis. In this review, we focused on severe lung pneumonia, a respiratory infection that affects the lower respiratory tract. It can be caused by various microorganisms, including viruses (e.g., influenza, SARS-CoV-2, RSV) and bacteria (e.g., streptococcus pneumoniae). Bacterial pneumonia is the deadliest infection, with a mortality rate approaching 20% among patients older than 85 years [44].

Chemerin levels were assessed in several sepsis and lung infection conditions. Regarding COVID-19 infection, one group reported higher serum chemerin concentrations in healthy controls compared to COVID-19 patients [45]. In a second study, the same group did not highlight significant differences in serum chemerin levels between non-severe and severe COVID-19 patients [46]. They observed a decrease in chemerin concentrations between days 1 and 7, followed by an increase between days 7 and 28 in moderate and severe patients, with a continuous increase in mild patients [46]. The authors did not find any correlation between chemerin levels and inflammatory biomarkers and hypothesized that the increase in serum chemerin concentrations was associated with the resolution of inflammation. However, chemerin levels observed in these two studies were significantly different, in the pg/mL range when measured with multiplex in Sulicka-Grodzicka et al. [46] and in the ng/mL range when assessed with ELISA in Kukla et al. [45], making it difficult to interpret their data (Figure 3). Esendagli et al. also measured the chemerin serum concentration of COVID-19 patients at hospital admission and separated them according to their prognosis, which depended on the type of ventilation and survival. Patients with a good prognosis presented higher chemerin levels at admission compared to patients with a bad prognosis; however, no significant difference was observed if patients had lung sequellae or not [47].

**Table 1 cells-13-00171-t001:** General overview of results obtained regarding the pro- or anti-inflammatory role of chemerin in human, in vivo and in vitro studies.

Diseases	Human Study	In Vivo Study	In Vitro Study
**Acute lung inflammation and sepsis**	Amend et al., Karampela et al., Horn et al.: higher chemerin levels (plasma and serum) in septic patients compared to controls [48,49,50]. Karampela et al., Horn et al.: association between chemerin levels and severity of the sepsis [49,50]. Ebihara et al.: no difference in chemerin levels between septic patients and controls [51].	Luangsay et al.: anti-inflammatory properties of the chemerin/CMKLR1 axis in the LPS model [35]. Provoost et al.: pro-inflammatory properties in the DEP model [37]. Luangsay et al., Malik et al.: effects of chemerin mediated only by CMKLR1 and not by CCRL2 [35,40]. Luangsay et al., Provoost et al., Razvi et al., Malik et al., Zou et al.: elevation of chemerin (BALF and lung) in all models of acute lung inflammation [35,37,39,40,41]. Horn et al.: higher blood chemerin concentrations in mice with severe septic shock [50].	Bondue et al.: no synergistic effect of chemerin on stimulation of pro- or anti-inflammatory cytokines’ secretion by activated macrophages [52]. Cash et al.: inhibition of secretion of pro-inflammatory cytokines and stimulation of secretion of anti-inflammatory cytokines by macrophages induced by picomolar concentrations of chemerin [53].
**Lung infection**	Kukla et al.: lower serum chemerin levels in COVID-19 patients compared to controls [45]. Lavis et al., Amend et al.: higher plasma chemerin levels in severe COVID-19 patients compared to controls [48,54]. Sulicka-Grodzicka et al.: no difference in serum chemerin levels between severe and non-severe patients [46]. Esendagli et al.: higher serum chemerin levels in COVID-19 patients with good prognosis compared to patients with bad prognosis [47]. Lavis et al.: higher plasma chemerin levels in deceased COVID-19 patients compared to recovered, and independent risk factor for mortality [54].	Bondue et al.: anti-inflammatory properties of the chemerin/CMKLR1 axis in a model of severe lung pneumonia, mediated by non-leucocytic cells [55].	Shirato et al.: decreased viral replication in A549 cells inactivated for *RARRES2* [56].
**Asthma**	Zhou et al.: higher plasma chemerin levels in asthmatic patients compared to controls [57].	Provoost et al., Zhao L. et al.: anti-inflammatory properties of chemerin in asthmatic mouse models induced by house dust mite and DEP and by ovalbumin [37,58].	Zhao L. et al.: inhibition of CCL2 secretion by primary lung epithelial cells if exposed to chemerin [58].
**Chronic obstructive pulmonary disease (COPD)**	Boyuk et al., Li C. et al.: higher chemerin levels (serum and plasma) in COPD patients [59,60]. Li C. et al.: association between plasma chemerin levels and disease severity (hospitalizations) [60]. Galecka et al.: no significant difference in serum chemerin levels between COPD patients and controls [61].	Demoor et al.: pro-inflammatory properties of the chemerin/CMKLR1 axis in COPD mouse model induced by the subacute and chronic exposure to tobacco smoke [62].	Absence of in vitro study regarding chemerin and COPD.
**Systemic sclerosis (SSc)**	Sawicka et al.: higher serum chemerin concentrations in SSc patients compared to controls [63]. Akamata et al.: no difference in serum chemerin concentrations between SSc patients and controls [64]. Chighizola et al.: lower serum chemerin concentrations in SSc patients compared to controls [65]. Sanges et al.: higher serum chemerin concentrations in SSc patients with PAH compared to SSc patients without PAH. Upregulation of the chemerin/CMKLR1 axis in lung vessels from PAH-SSc patients [66]. Peng et al.: higher plasma chemerin levels in PAH patients compared to controls [67]. Saygin et al.: upregulation of chemerin in fibroblasts from patients with idiopathic PAH [68].	Omori et al., Peng et al.: upregulation of CMKLR1 expression in lungs and of chemerin expression in plasma and lungs in PAH rats [67,69].	Omori et al.: contraction of isolated pulmonary arteries induced by chemerin and greater effect on arteries isolated from PAH rats [69]. Hanthazi et al.: potentiation of vasoconstrictor effects and antagonization of vasodilatator effects by chemerin [70]. Peng et al.: upregulation of chemerin and CMKLR1 expression by isolated smooth muscle cells if exposed to recombinant chemerin or hypoxia. Migration and proliferation of these cells by chemerin [67].
**Lung cancer (NSCLC)**	Sotiropoulos et al., Xu et al.: higher serum chemerin levels in patients with NSCLC compared to controls but controversy over the association between chemerin levels, lymph node involvement and tumoral stage [71,72]. Li F. et al.: no difference in serum chemerin levels between NSCLC patients and controls [73]. Zhao S. et al., Cai et al.: association between increased chemerin expression by tumor cells in lung slides from patients with NSCLC and good prognosis [74,75]. Zhao H. et al.: association between higher expression of RARRES2 in patients with NSCLC and good prognosis [76].	No mouse model studying chemerin and lung cancer but other studies showing anti-tumoral properties of chemerin. Pachynski et al.: increased recruitment of NK and T cells in a mouse model of melanoma tumor cells overexpressing chemerin leading to smaller tumors [77]. Al Delbany et al.: decreased neoangiogenesis in a chemical model of mouse skin carcinogenesis leading to smaller tumors [27]. Dubois-Vedrenne et al.: involvement of chemerin only in latter stages of tumorigenesis [78].	Controversy over the anti- or pro-tumoral role of chemerin.

These results are contradictory to the ones obtained by our group. Indeed, we observed higher chemerin plasmatic levels in COVID-19 patients compared to healthy controls, and patients hospitalized in the intensive care unit had the highest levels [54]. This is in agreement with previously described results showing a positive correlation between chemerin plasma concentration and inflammatory biomarkers [79]. Moreover, we demonstrated that chemerin concentrations on day 14 of hospitalization were an independent risk factor of mortality. We also showed by the histological examination of lungs from autopsied COVID-19 patients that chemerin is mainly expressed by fibroblasts/myofibroblasts in fibrotic lesions of diffuse alveolar damage [54].

Our results are in agreement with measurements of chemerin concentration obtained in septic patients. Indeed, Amend et al. observed higher chemerin plasma levels in COVID-19 patients compared to healthy controls. Of note, chemerin levels were identical between septic patients secondary to SARS-CoV-2 infection and other causes of sepsis, and no significant difference was observed between deceased or recovered patients. They also observed a positive and moderate correlation between chemerin levels and the C-reactive protein [48]. Concordant results were also obtained by Karampela et al. In this study, chemerin serum concentrations were higher at admission in septic patients compared to controls. The concentration was also higher in patients with septic shock, and chemerin levels could even discriminate patients with sepsis from patients with septic shock with a sensitivity and a specificity of approximately 70%. Chemerin levels were higher in deceased patients compared to recovered patients, and at days 1 and 7 of admission they were independent risk factors of mortality [49]. Similar conclusions were described by Horn et al., who observed higher chemerin levels in septic patients compared to controls, with higher levels in patients with a higher severity score for sepsis. This was also demonstrated in a mouse model of peritoneal septic shock, where mice with severe septic shock had the highest chemerin concentration [50]. Finally, one study did not observe any significant difference in serum chemerin levels between septic patients compared to controls, nor any difference according to mortality, but their cohort only included 37 septic patients and 12 controls [51].

Besides those observational studies, only a few studies have evaluated the role of the chemerin/CMKLR1 system in the pathophysiology of pneumonia. In a model of severe viral pneumonia induced by the pneumonia virus of mice, the mice counterpart of the respiratory syncytial virus (RSV), an anti-inflammatory role of the system was demonstrated. Indeed, CMKLR1^KO^ mice presented a more severe disease with higher mortality, lung inflammation and viral titers. BALF chemerin levels were also increased. Anti-viral cytokines such as interferon-α were strongly decreased, while pro-inflammatory cytokines rose sharply. The increase in inflammatory cells was accompanied by a significant decrease in plasmacytoid DC and CD8^+^ T cells. Additional experiments suggested that the anti-inflammatory properties of the chemerin/CMKLR1 system depended on non-leucocytic cells [55]. In contrast, in vitro experiments showed that inactivation of *RARRES2* decreased the sensibility of an alveolar basal epithelial cell line (A549 cells) to RSV infection, as observed by decreased viral replication [56]. The in vitro results could be explained by the lack of interference of the immune response to RSV infection. While chemerin may have a direct pro-viral role in lung cells, the anti-viral effects of this chemokine in the immune system may prevail, resulting in the beneficial effect observed in the in vivo model.

### 3.3. Chemerin and Obstructive Pulmonary Diseases

Asthma and chronic obstructive pulmonary disease (COPD) represent the most frequent chronic respiratory diseases, affecting around 7% of the population [80]. Both are obstructive pulmonary diseases, generally characterized by their reversible (asthma) or irreversible (COPD) nature. The immune response observed in asthmatic patients is mostly mediated by T helper 2 (Th2) cells which lead to an infiltration by eosinophils. Mast cells are also involved in the secretion of bronchoconstrictive mediators [81,82]. COPD develops as a result of repeated injuries to the airway epithelium driving a reprogramming of basal cells. The main immune cells involved in the pathogenesis of COPD are neutrophils and macrophages, but their precise roles are not well understood yet [83].

It has been previously shown that plasma chemerin concentrations were higher in patients with severe asthma compared to healthy controls [57]. A mouse model of allergic asthma induced by exposing the mice to ovalbumin highlighted the role of chemerin in the pathophysiology of this disease. Indeed, the addition of recombinant chemerin to ovalbumin led to a decrease in BALF immune cells, notably eosinophils, inflammatory DC and CD4^+^ T cells, along with a decrease in IL-4 and IL-13 BALF concentrations. The number of goblet cells was also reduced. The anti-inflammatory properties of chemerin were linked to a direct interaction with lung epithelial cells, leading to a decrease in CCL2 secretion, a chemoattractant agent for inflammatory DC, and not by a direct role on these immune cells [58]. Similar anti-inflammatory properties of chemerin were observed in a mouse model of aggravated allergic airway inflammation induced by simultaneously exposing mice to DEP and house dust mites. CMKLR1^KO^ mice presented a higher recruitment of monocytes, neutrophils, eosinophils, DCs and T cells compared to WT mice [37].

Regarding the link between chemerin and COPD, an exhaustive review was recently published and only main results will be presented here [84]. Two independent studies reported higher chemerin levels in COPD patients compared to healthy controls [59,60]. While Li et al. observed a higher chemerin concentration in hospitalized COPD patients and a positive correlation between chemerin concentrations and number of hospitalizations over 6 months [60], Boyuk et al. could not show any difference in chemerin levels according to the severity of COPD [59]. Additionally, a third independent study did not demonstrate any difference in serum chemerin levels between COPD patients and matched healthy controls or any correlation with respiratory function [61].

In order to study the pathophysiological role of chemerin in COPD, WT and CMKLR1^KO^ mice were exposed to tobacco smoke for 4 weeks (subacute exposure) or 24 weeks (chronic exposure). Both subacute and chronic exposure led to a decrease in preprochemerin and CMKLR1 expression in lungs from WT mice. However, only chronic exposure significantly decreased chemerin expression in the lung bronchial epithelium, whereas no effect for subacute exposure was observed. This decrease was associated with an increase in chemerin concentration in the BALF, suggesting, as previously described [37], a release of chemerin by epithelial cells. In this model, CMKLR1^KO^ mice were partially protected against the subacute inflammatory response, with a lower recruitment of immune cells and reduced secretion of chemokines, showing a proinflammatory role for the chemerin/CMKLR1 axis in this model [62].

### 3.4. Chemerin and Autoimmune Diseases

Autoimmune diseases are caused by deregulated responses of the immune system, leading to damage of self-tissue and disruption of its normal function.

Chemerin was mostly characterized in systemic sclerosis (SSc), an autoimmune connective disease, leading to fibrosis of the skin and internal organs [85]. Lungs can be affected during the course of the disease and their involvement is responsible for the death of most of SSc patients [86]. The main pulmonary lesions observed are interstitial lung disease (ILD) and pulmonary hypertension (PAH).

Measurements of chemerin serum concentration in patients with SSc gave contradictory results and were dependent on the population of SSc patients studied. Thus, either higher [63], no difference [64] and even lower chemerin concentrations [65] were observed in SSc patients as compared to controls. Recently, Sanges et al. compared SSc patients with or without PAH and observed higher levels of chemerin in serum from PAH-SSc patients. Chemerin levels in PAH-SSc patients were also higher compared to SSc patients with ILD but without PAH. They also observed an upregulation of the chemerin/CMKLR1 axis in the lung vessels of PAH-SSc patients compared to healthy controls [66]. On the other hand, a correlation between PAH and chemerin levels was not found by Sawicka et al., maybe due to their smaller patient cohort [63].

As chemerin and CMKLR1 are expressed by endothelial cells and CMKLR1 by smooth muscle cells, the link between the chemerin/CMKLR1 axis and PAH was investigated. In vitro studies showed that chemerin 9, the shortest chemerin-derived peptide that retains the highest potency against CMKLR1 [11], could induce a contraction of an isolated pulmonary artery and that the effect was greater when the arteries were isolated from rats with PAH. They also observed an increase in CMKLR1 and a decrease in CCRL2 expression in lungs from PAH rats as compared to control rats, as well as an increase in chemerin expression in plasma [69]. It was also demonstrated that recombinant chemerin potentiated the effects of vasoconstrictors such as phenylephedrine, endothelin 1 and serotonin and antagonized the vasodilatation effect of acetylcholine. In opposition to this study, it was demonstrated that recombinant chemerin led to pulmonary artery vascular contraction only in the absence of endothelium [70]. The same results were obtained by Peng et al., who observed an overexpression of chemerin and CMKLR1 in lungs from rats with PAH. Smooth muscle cells isolated from the pulmonary artery and treated with recombinant chemerin or exposed to hypoxia displayed a higher expression of chemerin and CMKLR1, suggesting an autocrine role for the system. Chemerin also favored the migration and proliferation of these cells. Moreover, the concentration of chemerin was significantly increased in plasma from idiopathic PAH patients, and a chemerin concentration above 471.76 pg/mL could predict a PAH diagnosis with a sensitivity of 85.7% and a specificity of 100% [67]. Explants from patients with idiopathic PAH were also analyzed and showed an upregulation of chemerin in fibroblasts from idiopathic PAH compared to controls [68].

Regarding rheumatoid arthritis, even if chemerin seems to be implicated in the physiopathology of the disease, no study has evaluated the link between chemerin and the pulmonary lesions associated to it [87].

### 3.5. Chemerin and Lung Cancer

The link between inflammation and cancer is a rapidly growing research area, and chemerin has been studied in cancer, and notably in lung cancer, as it was hypothesized that its chemoattractant properties could mediate the recruitment of tumor-associated immune cells and influence neoangiogenesis [88].

Zhao et al. found a seven-gene panel, including *RARRES2*, that was associated with the lung tumoral microenvironment. Patients with a higher expression of *RARRES2* had a better overall survival. Using the seven-gene panel, they accurately divided patients into low risk and high risk of progression or death [76]. In line with this study, immunohistochemistry analysis of non-small cell lung carcinoma (NSCLC) revealed that chemerin expression was decreased in tumor cells compared to the normal adjacent stroma, and the more the tumor was differentiated, the more chemerin was expressed. Patients with higher chemerin expression had a better survival, and the expression of chemerin was demonstrated as an independent risk factor for five-year progression-free survival [74]. Similar results were obtained by Cai et al. [75]. Two independent teams assessed chemerin concentrations in serum from a large cohort of patients with NSCLC. They both observed higher serum chemerin levels in NSCLC patients and could discriminate NSCLC patients from healthy controls based on chemerin concentrations with a sensibility of around 63%. However, the cut-off of chemerin values differed strongly between the two cohorts, and contradictory results were obtained regarding the association between chemerin concentrations, lymph node involvement and tumoral stage. Of note, one cohort comprised only resectable NSCLC (absence of metastasis) and the other comprised around 30% of metastatic patients [71,72]. Only one smaller study did not find a significant difference in chemerin concentrations in serum in NSCLC patients compared to controls [73].

The mechanisms through which chemerin influences lung cancer have not been thoroughly investigated. However, certain in vivo models have attempted to unravel the connection between chemerin and the development of cancer. Mice inoculated with B16 melanoma tumor cells overexpressing chemerin had smaller tumors with an enhanced recruitment of NK and T cells [77]. It was also shown in the same disease model using CMKLR1^KO^ and CCLR2^KO^ mice that the decrease in the tumor size was mostly due to decreased tumor neoangiogenesis induced by chemerin, with no remarkable changes in lymphocyte infiltration [27]. Dubois-Vedrenne et al. observed in a model of chemical skin tumors that mice overexpressing a bioactive form of chemerin developed fewer and smaller tumors. These effects were partially mediated by CMKLR1 and were only linked to the action of chemerin in the latter stages of tumorigenesis [78]. Of note, the pro-tumoral effects of chemerin were only observed in in vitro models, with enhanced migration properties in models of squamous cell carcinoma [89,90]. In vitro models do not provide a comprehensive understanding of crucial cellular interactions, particularly those occurring between cancer cells and the tumor microenvironment. Given that the anti-tumoral effects of chemerin appear to be closely associated with responses from immune and endothelial cells, prioritizing in vivo studies becomes imperative.

## 4. Conclusions

Despite chemerin having a validated role in inflammation, the precise mechanisms that determine the balance between its anti- and pro-inflammatory properties have not been fully clarified. A common hypothesis for its dual role originates from its various isoforms that can be obtained depending on the C-terminal extremity cleavage. As mentioned above, the most active forms of chemerin (21-156 and 21-157) come from cleavage by proteases produced by neutrophils, the dominant cells in acute inflammation. These forms of chemerin are recognized as major chemoattractant agents for immune cells, assuming a pro-inflammatory role for the protein. In asthma models, an anti-inflammatory role of chemerin was observed and could partially be explained by the enhanced presence of mast cells that are responsible for the production of inactive chemerin 21-154 [16,37,58]. A balance between chemerin isoforms depending on the cells present could then promote a varying degree of cell recruitment. Unfortunately, most of the studies conducted do not differentiate between prochemerin and active chemerin and fail to identify the various forms of chemerin present. This limitation is attributed to the need for high-performance liquid chromatography to determine the different isoforms, followed by an aequorin assay to establish their specific activity. Conducting these experiments requires a significant amount of material for analysis and a high level of expertise [78]. The quantity of active chemerin present could also influence the recruitment of particular immune cells. Indeed, similar to other chemoattractant agents, an optimal chemerin concentration is needed for optimal cell migration, and this concentration differs between cell types. Our group observed that the optimal concentration is around 3 nM for DC migration and around 1 nM for macrophages [35]. Depending on the quantity of prochemerin released by non-leucocytic cells and the quantity of active forms generated, the recruitment of some immune cells could be favored and lead to pro- or anti-inflammatory effects. This could therefore explain the differences observed in the two models of acute lung inflammation—the anti-inflammatory role in the LPS model leading to a greater acute inflammation than the DEP exposure model, causing a less severe lung inflammation and showing a pro-inflammatory role of the chemerin/CMKLR1 system [35,37].

The direct role of chemerin in the secretion of pro- or anti-inflammatory cytokines by immune cells is controversial. Indeed, our group previously showed that the in vitro activation of murine peritoneal macrophages with LPS and interferon-ɣ associated with recombinant chemerin did not increase the secretion of pro-inflammatory cytokines (IL-6, IL-1 and TNF-α) or anti-inflammatory cytokines (IL-10). The same results were obtained on stimulated human macrophages [52]. However, Cash et al. observed that picomolar concentrations of chemerin inhibited the secretion of pro-inflammatory cytokines (IL-1β, IL-6, IL-12 p40, RANTES and TNF-α) and stimulated the secretion of anti-inflammatory cytokines (IL-10) by macrophages [53]. New independent studies should be carried out to obtain a definitive answer.

Some studies have also described that the anti-inflammatory properties of chemerin depended on non-leucocytic cells, notably the lung epithelial cells. In the asthma model induced by ovalbumin, the anti-inflammatory role of chemerin was not linked to a direct effect on the recruitment of pro-inflammatory DCs but to an interaction with lung epithelial cells, decreasing their secretion of CCL2, another chemoattractant agent [58]. The model of severe viral pneumonia also demonstrated that the anti-inflammatory properties of chemerin were linked to non-leucocytic cells but without being able to precisely identify the cell(s) type(s) involved [55].

As previously mentioned, most studies have shown that chemerin concentrations are elevated in diseases associated with lung inflammation (Figure 4). However, there is considerable variability in chemerin levels across different studies, which are probably dependent on the type of assay used and the origin of the chemerin (plasma or serum; Figure 3). As an example, the same group obtained chemerin concentrations in the pg/mL range using a multiplex immunofluorescence assay or in the ng/mL range using an enzyme-linked immunosorbent assay. Additionally, there are no established guidelines regarding the preferred blood sample (plasma or serum), storage or assay to be used, leading research teams to use either without clarity on potential results variation. Although chemerin appears to be a promising biomarker for numerous inflammation-related lung diseases, standardizing measurement techniques is crucial to obtain reproducible results across different centers and facilitate its implementation in routine clinical practice.

In summary, deciphering the mechanisms that underline the pro- and anti-inflammatory properties of chemerin is crucial to understanding how this chemokine exerts its effects. The identification of chemerin isoforms in inflammatory diseases, as well as the interactions between chemerin, non-leucocytic cells and immune cells, should be further investigated.

## Figures and Tables

**Figure 1 cells-13-00171-f001:**
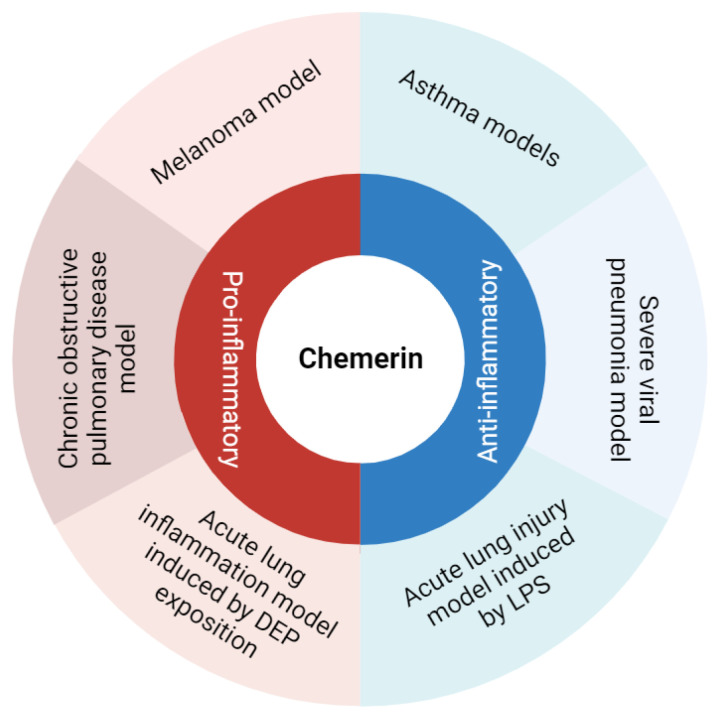
Summary of chemerin pro- and anti-inflammatory roles observed in various models of lung diseases. Figure created with Biorender.com.

**Figure 2 cells-13-00171-f002:**
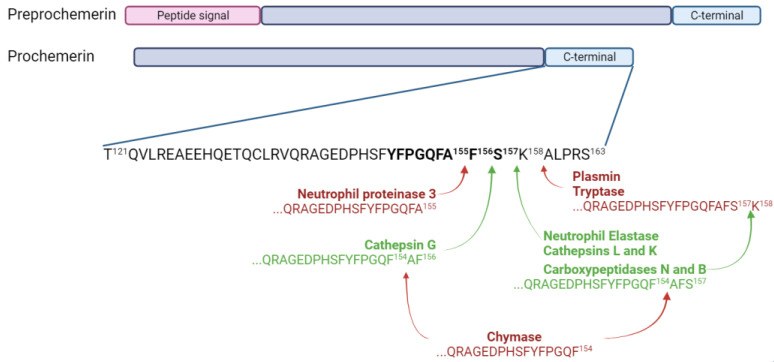
Chemerin and its isoforms. Chemerin 9, the smallest peptide retaining significant bioactivity, is shown in bold. Enzymes producing inactive forms of chemerin are in bold red and the ones producing active forms are in bold green. Figure created with Biorender.com.

**Figure 3 cells-13-00171-f003:**
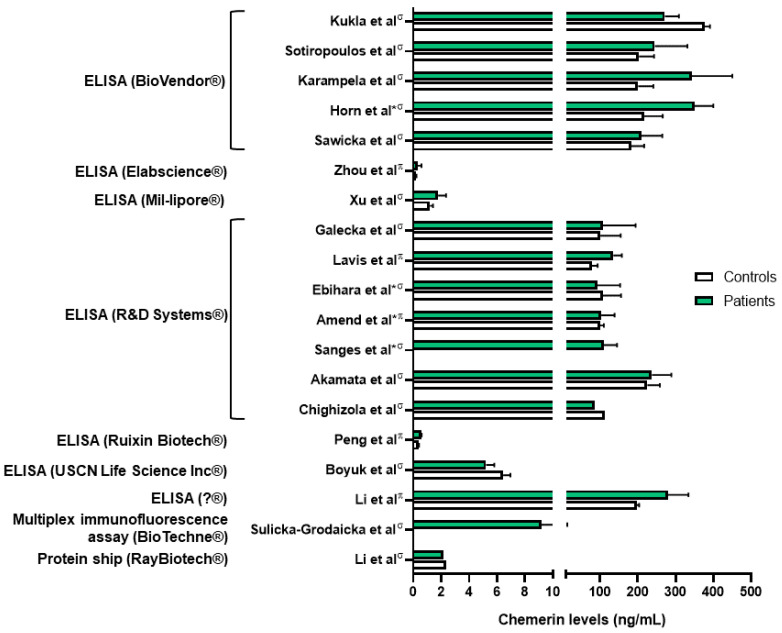
Chemerin levels measured in different studies from patients with various lung diseases (green column bars) and controls (white column bars). ^σ^: measures performed in serum samples; ^π^: measures performed in plasma samples; *: chemerin levels extrapolated from graphs. Data from [45,48,49,50,51,54,57,59,60,61,63,64,65,66,67,71,72,73]. Figure created with Biorender.com.

**Figure 4 cells-13-00171-f004:**
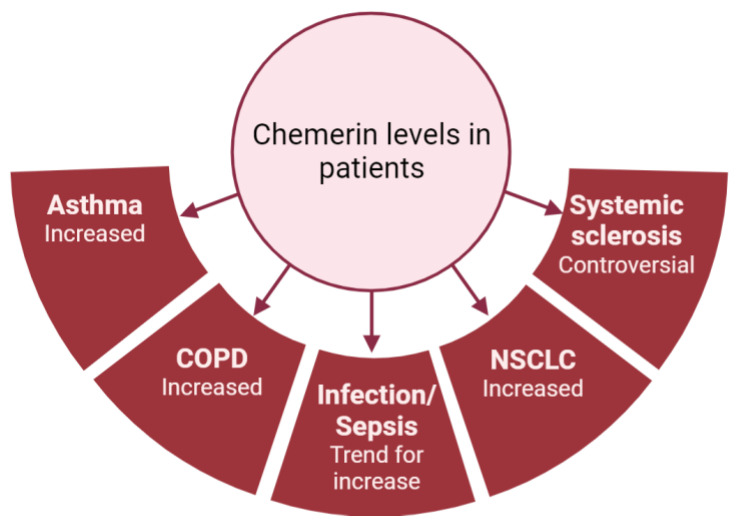
Summary of chemerin levels obtained in different studies from patients with various lung diseases compared to controls. Figure created with Biorender.com.

## Data Availability

Not applicable.

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
