# Peer review of "The Dual Role of Chemerin in Lung Diseases"

_cells, 2024, doi:10.3390/cells13020171_

Round 1
Reviewer 1 Report
Comments and Suggestions for Authors
After a thorough analysis, I recommend the article for publication with minor corrections, which basically only concern the unification of the literature, in line with the requirements for the authors.
Please harmonize the literature in accordance with the requirements for authors:
Item no [4] – Insert the abbreviation of the journal
Item no [11] – remove dots
Item no [24] – remove the dot at the end of the journal name
Item no [25] – Insert the abbreviation of the journal Int J Mol Sci
Item no [25] – remove dots
Item no [75] – Insert the abbreviation of the journal
Item no [77] – Insert the abbreviation of the journal
Item no [89] – PLOS One
The numbering of the reference list should be in square brackets
Author Response
We thank the reviewer for his positive feedback on our manuscript. We corrected the list of references as required by the journal.
Please see complete answers to the reviewers in the attached document.

Reviewer 2 Report
Comments and Suggestions for Authors
This is a well written and well organized review article described the relations of chemerin and different types of lung diseases, such as ARDS, lung infection, COPD, ILD, and lung cancer. However, the title is about the lung inflammation, though all the diseases are associated with inflammation, considering revising the title to cover a bit broader might be better to avoid misleading of the topic.
In the following paragraph of introduction, the authors described the role of chemerin in lung disease a figure or a table for an overview of the expression level of chemerin in patients accross all disease types mentioned in this review will be good for the audience to understand it better.
For example, figure2 . But all different results from different articles are not needed in my point of view . An overview will be enough to describe one associated disease to chemerin.
Final, can the author list a table describe the results from human subject vs animal study vs cell model? That will be very helpful!
Author Response
Please see in attachment the complete answers to the reviewers and editor.

Reviewer 3 Report
Comments and Suggestions for Authors
The authors of the manuscript titled: The dual role of chemerin in lung inflammation have comprehensively reviewed the literature available on the pro- and anti-inflammatory properties of this chemokine in various inflammatory lung conditions. In addition, the authors discuss the potential use of chemerin as a therapeutical target due to its immunomodulatory role. From their review of the literature, it is clear that the type of sample and the technique used to measure the levels of chimerin needs to be standardised before it can be considered as a potential biomarker for the diagnosis and/or the prognosis of inflammatory lung diseases.
The authors have used current references and, although they do reference their own work, this is appropriate. The review is extensive and no area is lacking in detail and information. The authors have made appropriate use of figures to highlight their findings of the current literature.
This review is timely and appropriate for inclusion in the journal: Cells.
Comments on the Quality of English LanguageThe manuscript does need to be read and corrected by someone proficient in English.
The use of certain words needs to be corrected. In particular, the use of 'exposition' needs to be corrected to 'exposed to'.
Author Response
We express our gratitude to the reviewer for their positive assessment of our article. In compliance with the reviewer's suggestions, we have thoroughly proofread and rectified our work for vocabulary, grammatical, and syntactical errors.
Please see in attachment the complete answers to the reviewers and editor.
